# Progress in Oligometastatic Prostate Cancer: Emerging Imaging Innovations and Therapeutic Approaches

**DOI:** 10.3390/cancers16030507

**Published:** 2024-01-24

**Authors:** Ryo Oka, Takanobu Utsumi, Takahide Noro, Yuta Suzuki, Shota Iijima, Yuka Sugizaki, Takatoshi Somoto, Seiji Kato, Takumi Endo, Naoto Kamiya, Hiroyoshi Suzuki

**Affiliations:** Department of Urology, Toho University Sakura Medical Center, 564-1 Shimoshizu, Sakura-shi 285-8741, Chiba, Japan; ryou.oka@med.toho-u.ac.jp (R.O.); takahide.noro@med.toho-u.ac.jp (T.N.); yuta.suzuki@med.toho-u.ac.jp (Y.S.); shouta.iijima@med.toho-u.ac.jp (S.I.); yuuka.kizuki@med.toho-u.ac.jp (Y.S.); takatoshi.soumoto@med.toho-u.ac.jp (T.S.); seiji.katou@med.toho-u.ac.jp (S.K.); takumi.endou@med.toho-u.ac.jp (T.E.); naoto.kamiya@med.toho-u.ac.jp (N.K.); hiroyoshi.suzuki@med.toho-u.ac.jp (H.S.)

**Keywords:** cytoreductive surgery, metastasis-directed therapy, oligometastatic prostate cancer, PSMA-PET, stereotactic body radiation therapy

## Abstract

**Simple Summary:**

In recent years, there has been growing interest in oligometastatic prostate cancer (PCa). Unlike more widespread forms of the disease, oligometastatic PCa involves only a limited number of cancerous lesions in specific areas of the body. This review delves into the latest advancements in our understanding of oligometastatic PCa, including how it works on a biological level, the use of advanced imaging techniques to spot it, and the various treatment approaches being explored. For patients with this intermediate-stage cancer, there is hope on the horizon, as personalized treatments such as surgery and targeted radiation therapy are showing promise. The findings from ongoing research may ultimately lead to better outcomes for individuals facing this unique challenge, bridging the gap between localized and widespread PCa.

**Abstract:**

Prostate cancer (PCa) exhibits a spectrum of heterogeneity, from indolent to highly aggressive forms, with approximately 10–20% of patients experiencing metastatic PCa. Oligometastatic PCa, characterized by a limited number of metastatic lesions in specific anatomical locations, has gained attention due to advanced imaging modalities. Although patients with metastatic PCa typically receive systemic therapy, personalized treatment approaches for oligometastatic PCa are emerging, including surgical and radiotherapeutic interventions. This comprehensive review explores the latest developments in the field of oligometastatic PCa, including its biological mechanisms, advanced imaging techniques, and relevant clinical studies. Oligometastatic PCa is distinct from widespread metastases and presents challenges in patient classification. Imaging plays a crucial role in identifying and characterizing oligometastatic lesions, with new techniques such as prostate-specific membrane antigen positron emission tomography demonstrating a remarkable efficacy. The management strategies encompass cytoreductive surgery, radiotherapy targeting the primary tumor, and metastasis-directed therapy for recurrent lesions. Ongoing clinical trials are evaluating the effectiveness of these approaches. Oligometastatic PCa occupies a unique position between locally advanced and high-volume metastatic diseases. While a universally accepted definition and standardized diagnostic criteria are still evolving, emerging imaging technologies and therapeutic strategies hold promise for improving the patient outcomes in this intermediate stage of PCa.

## 1. Introduction

Prostate cancer (PCa) stands as the second most prevalent malignancy and the primary cause of cancer-related mortality within the male population worldwide [1,2]. The incidence and mortality rates of PCa are exhibiting an upward trajectory, influenced by age-associated trends in oncogenesis across numerous countries, despite advances in diagnostic and therapeutic methodologies [3,4]. PCa encompasses a spectrum of heterogeneity, ranging from indolent to highly aggressive phenotypes. Approximately 10 to 20% of all PCa patients are afflicted by metastatic PCa, with the recent trends showing an increasing incidence [1,5,6,7]. Oligometastatic PCa has garnered heightened attention, representing an initial phase in the metastatic progression of cancer dissemination, colonization, expansion, and ongoing interactions, thanks to the advent of newer imaging modalities [8,9]. Oligometastatic PCa is distinguished by the presence of a limited number of metastatic lesions in specific anatomical locations [8]. The prospect of achieving complete remission in oligometastatic PCa has engendered debates concerning the appropriate selection of therapeutic strategies [10].

Patients with metastatic PCa typically receive systemic therapy, such as androgen deprivation therapy (ADT), in conjunction with androgen-receptor-axis-targeted therapies (ARATs) and/or chemotherapy, contingent upon the extent of the metastatic tumor burden [11,12]. In recent years, there has been a burgeoning demand for personalized treatment approaches for metastatic PCa. Notably, for oligometastatic PCa characterized by only a limited number of metastatic lesions, the potential utility of surgical interventions and radiotherapy (RT) at local and/or metastatic sites has gained prominence. Several prospective clinical trials focusing on oligometastatic PCa are currently underway [13,14,15,16].

Within this comprehensive review, we provide a consolidated overview of the latest advancements pertaining to oligometastatic PCa, delving into the investigations concerning its underlying biological mechanisms, the impact of state-of-the-art imaging techniques, and a discussion of pertinent clinical studies in the realm of oligometastatic PCa.

## 2. Definition of Oligometastasis

Oligometastasis, as introduced by Hellman and Weichselbaum in 1995, is a concept postulating that patients with metastatic cancer harboring a limited number of metastases and confined sites of metastasis might achieve curative outcomes through the comprehensive management of all metastatic lesions using RT or surgical interventions [8]. Oligometastasis is delineated by various contributing factors, including: (1) the cancer’s inherent sluggish growth, (2) the cancer’s modest metastatic propensity, (3) an early diagnosis during the metastatic course of a rapidly proliferating cancer, and (4) the heightened detectability of metastatic foci owing to advancements in imaging modalities [17,18]. Oligometastasis is fundamentally characterized by the presence of slowly evolving metastatic lesions that manifest on a monthly to yearly basis. However, due to the challenge of distinguishing between indolent and aggressive cancers at the time of an oligometastasis diagnosis in clinical practice, certain patients may harbor rapidly expanding micrometastases that become evident through imaging assessments [14,15]. Initially, oligometastasis did not encompass the elimination of uncontrolled primary sites accompanied by multiple distant metastases. Subsequently, Niibe et al. introduced the concept of oligorecurrence, which bears a resemblance to oligometastasis [19]. Oligorecurrence entails controlled primary cancer sites, allowing for the treatment of all gross recurrent or metastatic regions through localized therapeutic approaches, leading to improved post-treatment prognoses [19,20,21]. At present, oligometastasis has undergone a detailed classification based on previous research endeavors [8,19,20,21].

Oligometastatic PCa occupies an intermediate position between a locally advanced disease and high-volume metastases, necessitating differentiation in terms of the prognosis and therapeutic strategies [15]. Although a universally accepted definition of oligometastatic PCa and standardized diagnostic criteria are yet to be established, various criteria exist in clinical trials. In the majority of investigations, the threshold for the number of metastases is frequently set at less than 4–6, typically involving lymph node (N1 or M1a) or bone (M1b) sites [15,22]. In the Advanced Prostate Cancer Consensus Conference of 2019, 48% of the panelists advocated for a definition of less than four metastases as the threshold for oligometastatic PCa, while 42% proposed less than five metastases [23]. In terms of metastatic sites, 46% of the panelists advocated for lymph node and bone oligometastases (excluding visceral metastases), whereas 33% favored the inclusion of visceral metastases [23].

While the European Society for Radiotherapy and Oncology and the European Organisation for Research and Treatment of Cancer have recently introduced a classification system for oligometastases, categorizing them as de novo, repeat, or induced oligometastases [24], clinical practice simplifies the categorization of oligometastatic PCa into three distinct disease scales: de novo oligometastatic PCa, oligorecurrent PCa, and oligoprogressive PCa (Table 1) [9,10,12,23,25]. Castration-sensitive PCa (CSPC) with synchronous oligometastasis at the initial diagnosis is characterized as de novo, with synchronous oligometastasis remaining untreated in both metastatic and primary tumors. CSPC with metachronous oligometastasis following local therapy is designated as oligorecurrent, with metachronous oligorecurrence being untreated exclusively in metastatic foci. Castration-resistant PCa (CRPC) displaying metachronous oligometastasis subsequent to a systemic hormonal treatment is classified as oligoprogressive PCa, and oligoprogression warrants a comprehensive treatment encompassing both the primary and metastatic sites. Consequently, the treatment responses are likely to exhibit considerable disparities among these distinct scenarios.

## 3. Imaging Diagnosis of Oligometastatic Lesions

Conventional computed tomography (CT) and ^99m^Tc bone scintigraphy represent routine diagnostic tools for the evaluation of oligometastases in clinical practice. While CT and bone scintigraphy offer advantages in terms of cost-efficiency, practicality, and alignment with the established guidelines, their diagnostic sensitivity remains at approximately 70% to 80% for CT and 60–80% for bone scintigraphy [14,15,18]. Additionally, emerging imaging modalities, notably magnetic resonance imaging (MRI) and positron emission tomography (PET)-CT, have garnered attention for their utility in diagnosing oligometastasis [26,27]. Historically, MRI has found extensive applications in the assessment of localized prostate cancer and bone metastases, with the recent recognition of the value of whole-body MRI for the detection of PCa metastases [14,15,28].

In the realm of PET imaging, ^18^F-sodium fluoride (Na^18^F) stands as the internationally recognized radiotracer of choice, and it is particularly adept at identifying osteogenic metastases due to its avidity for bone remodeling sites. Na^18^F PET-CT surpasses bone scintigraphy in the detection of bone metastases [15]. Beyond Na^18^F, various radiotracers have exhibited efficacy, including 11C-choline and ^18^F-fluoroethylcholine for phospholipid synthesis, ^18^F-FACBC for protein synthesis, and ^68^Ga-prostate-specific membrane antigen (PSMA)-11 and ^18^F-DCFPyL for PSMA imaging [14,15,26,27,28,29,30]. Earlier, ^18^F-FACBC PET-CT and MRI were deemed proficient in the identification of metastatic lesions among CRPC patients [29,30]. Notably, PSMA-PET, encompassing ^68^Ga-PSMA-11 and ^18^F-DCFPyL, presently stands as the preeminent imaging modality tailored to prostate cancer. PSMA, a transmembrane protein prominently expressed in prostate cancer cells, migrates from the inner prostate lumen to the epithelial surface [31]. PSMA-PET, in conjunction with a small molecule ligand binding to PSMA and a positron-emitting radioisotope, exhibits robust accumulation in PCa, affording high-contrast lesion detection. Remarkably, ^68^Ga-PSMA-11 PET-CT yielded positive results in 54% of patients experiencing biochemical recurrence following a radical prostatectomy (RP), with PSA levels below 1.0 ng/mL, despite conventional imaging failing to detect the metastasis [27]. Additionally, ^68^Ga-PSMA-11 PET-CT prompted a stage reclassification compared to conventional imaging in 62–76% of patients [32]. Furthermore, ^68^Ga-PSMA-11 PET-CT has demonstrated exceptional utility in identifying oligometastases, and is characterized by its high interobserver reproducibility. Presently, the establishment of a conclusive link between the identification of oligometastatic afflictions through new-generation imaging and advancements in survival consequences remains elusive. Nevertheless, these emerging imaging modalities of the next generation, including PSMA-PET and whole-body MRI, are reshaping the framework and perspective surrounding oligometastasis, leading to potential stage reclassifications (Figure 1).

## 4. Treatment of the Primary and/or Metastatic Lesions

### 4.1. Rationale

The management of both the primary tumor and metastatic loci in oligometastatic PCa elicits cytoreductive effects, culminating in the diminished circulation of conventional and disseminated tumor cells. This attenuation results in a reduction in the interplay between the primary neoplasm and its metastatic counterparts, a process mediated by cytokines, chemokines, and microRNAs [15]. This diminishment in interplay has the potential to curtail the colonization of novel metastatic locales, abate the advancement of established metastases, and mitigate the reseeding of the primary site by malignant cells. Metastasis-directed therapy (MDT) may yield an abscopal phenomenon through immunomodulation, triggering a systemic anti-tumor response that exerts its influence on distant tumor foci [33]. Furthermore, the therapeutic intervention targeting the primary tumor may serve to forestall intratumoral adaptations and the emergence of castration resistance [34,35].

Colosini et al. conducted an initial investigation into the stratification of oligometastatic CSPC patients using a liquid biopsy. They enrolled 28 patients with oligometastatic CSPC who underwent stereotactic body radiation therapy (SBRT) as their therapeutic modality. They employed a deep targeted sequencing approach to analyze serum-derived cell-free DNA samples collected prior to the initiation of SBRT. Significantly, genetic mutations were detected in the genomic material of 25 out of the 28 patients, with the most prevalent alterations observed in genes such as ATM, BRCA1/2, and AR. The patients carrying BRCA1 mutations exhibited treatment failure following SBRT. In the future, the real-time molecular profiling of oligometastatic PCa may facilitate the identification of a genuine oligometastatic phenotype, thereby enabling a more favorable response to localized curative therapies or the attainment of sustained disease control over an extended period [36].

### 4.2. Radical Prostatectomy

The inclusion of cytoreductive surgery as an integral component of the therapeutic strategy for oligometastatic PCa has sparked debates due to concerns surrounding procedural complications versus its potentially marginal therapeutic efficacy. An RP for patients contending with oligometastatic PCa holds the potential for mitigating localized symptoms and complications, including hemorrhaging, bladder outlet obstruction, and ureteral obstruction, when contrasted with RT [37]. With the advent of robot-assisted laparoscopic RPs, which has substantially mitigated surgical invasiveness, cytoreductive surgery is now under consideration as a viable facet of treatment. This surgical approach shows promise for patients afflicted with oligometastatic PCa, encompassing both primary prostate lesions and metastatic foci residing in pelvic lymph nodes.

Prior retrospective investigations involving relatively modest case cohorts have indicated that cytoreductive surgery may extend the time to castration-resistant prostate cancer (CRPC), bolster the progression-free survival (PFS), enhance the cancer-specific survival (CSS), and ultimately confer a survival advantage in PCa patients harboring bone metastases [15,16]. In a comprehensive retrospective study utilizing data from the US Surveillance, Epidemiology, and End Results database spanning from 2004 to 2010, among 474 individuals with metastatic (M1a to M1c) PCa, those who underwent an RP exhibited significantly reduced cancer-specific mortality when compared to their non-local treatment counterparts (hazard ratio (HR) = 0.35 [95% confidence interval (CI): 0.35–0.46], *p* < 0.001) [35]. Furthermore, this benefit was more pronounced among relatively youthful patients [35]. However, it is imperative to acknowledge that these findings stem from retrospective analyses that encompassed patients beyond those with oligometastases, raising the possibility of variations in patient characteristics at the time of the RP [15,16,35].

To date, the utility of a cytoreductive RP for oligometastatic PCa remains unsupported by prospective randomized trials. In a case-control study focusing on oligometastatic PCa, 23 patients who underwent a cytoreductive RP in conjunction with preoperative ADT for low-volume bone metastases (defined as ≤3 metastases) experienced a significantly prolonged time to CRPC (40 months vs. 29 months, *p* = 0.04) and an improved PFS (38.6 months vs. 26.5 months, *p* = 0.03) in comparison to the 38 patients treated solely with ADT [38]. Conversely, in the sole prospective trial conducted by Steuber et al., which involved 43 patients with low-volume bone metastases (≤3 metastases), a cytoreductive RP lowered the risk of local complications, but did not confer statistically significant extensions in the overall survival (OS) or time to CRPC [39].

While the need for verification through prospective randomized trials is apparent, numerous clinical investigations are currently underway to explore the feasibility of randomization in de novo oligometastatic PCa [40]. Notably, the multi-institutional prospective randomized trial, denoted as the testing RP in men with PCa and oligometastases to the bone (TRoMbone) trial, as reported by Sooriakumaran et al., scrutinized the feasibility of RPs coupled with a pelvic lymphadenectomy in addition to the standard-of-care (SOC) regimen (ADT ± docetaxel) in newly diagnosed oligometastatic PCa patients [41]. This pioneering trial assessed primary and secondary endpoints concerning the feasibility of randomization, patient quality-of-life (QoL), and oncological outcomes and discovered that an RP for oligometastatic PCa patients was safe, yielding outcomes akin to surgery in standard indications. Notable rates of PSA (prostate-specific antigen)-PFS (PSA < 1 ng/mL) at 6 months post-surgery, Gleason scores of 8–10, a pT3 status, and positive margin rates were observed in 82.6%, 82.6%, 87.5%, and 41.7% of cases, respectively.

In the PRORAD trial (NCT03301701), PCa patients with fewer than six metastases (M1a/b/c, with no limits for N1) received ADT alongside SBRT as the MDT for metastatic lesions [42]. In Arm A, an RP was performed for the primary prostate, while Arm B received a high-dose rate of brachytherapy or SBRT. The primary endpoint was to assess the feasibility of randomization, with the secondary endpoints focused on the treatment efficacy and toxicity. Concurrently, the FUSCC-OMPCa trial (NCT02742675) is actively investigating the utility of RPs for primary tumors in patients with de novo oligometastatic PCa [40]. This trial is comparing ADT alone in Arm A with ADT in conjunction with local therapy (an RP or RT) in Arm B for de novo oligometastatic PCa patients harboring fewer than six oligometastases (N1 or M1a/b). The primary endpoint is centered on evaluating the PFS following a two-year timeframe. Should the results of the FUSCC-OMPCa trial affirm that ADT combined with local therapy can extend the PFS, an RP may emerge as a reinforced treatment option for de novo oligometastatic PCa. An RP for oligometastatic PCa appears to be associated with acceptable morbidity and safety profiles, making it particularly suitable for select patients.

### 4.3. Radiotherapy

RT is renowned for its capacity to induce the abscopal effect, leading to a reduction in or the eradication of distant lesions situated beyond the irradiated field. It is postulated that RT triggers local or systemic antitumor responses at the molecular level. Thus, the abscopal effect stands as a fundamental rationale for employing RT in the context of oligometastatic PCa. The application of RT in the management of oligometastatic PCa can be categorized into two main approaches: (1) targeting the primary tumor and (2) addressing metastatic lesions. Concerning the former approach, a prevailing method involves the selective irradiation of the primary tumor in cases of de novo oligometastatic PCa [43,44,45,46]. In contrast, the latter approach, termed MDT, entails the delivery of a curative dose through stereotactic body radiation therapy (SBRT) to all metastases that emerge subsequent to the radical treatment of the primary tumor with either an RP or RT [47,48]. Table 2 summarizes the clinical trials that have included RT for primary or metastatic lesions in oligometastatic PCa.

Subgroup analyses of the prospective randomized controlled trials (RCTs) HORRAD and STAMPEDE Arm H lend credence to the notion that RT directed at the primary tumor may confer benefits in the context of oligometastatic PCa [43,44]. The HORRAD trial, a multicenter RCT, encompassed 432 de novo metastatic CSPC (mCSPC) patients presenting with PSA levels exceeding 20 ng/dL and evidence of bone metastases on scans. The intervention arm underwent RT in conjunction with ADT, focusing on the prostate, while the pelvic lymph nodes were excluded. The control arm received ADT exclusively. The median OS was 45 months (95% CI: 40.4–49.6) in the RT group and 43 months (95% CI: 32.6–53.4) in the control group. Notably, no statistically significant differences in the OS were observed (HR: 0.90 [95% CI: 0.70–1.14], *p* = 0.4). However, a notable divergence in the 2-year survival emerged among patients with PSA levels < 142 ng/dL, <5 bone metastases, and Gleason scores < 8. A subgroup analysis within the cohort of patients with oligometastatic PCa harboring <5 bone metastases suggested a trend toward an improved OS in the ADT-with-RT group, albeit without statistical significance [43].

In the STAMPEDE Arm H trial, which enrolled 2061 de novo mCSPC patients, the objective was to compare the standard-of-care (SOC) group, consisting of ADT with or without docetaxel, with the SOC-plus-localized-prostate-external-beam-RT (SOC + RT) group [44]. Patients assigned to the RT group received either daily (55 Gy in twenty fractions over 4 weeks) or weekly (36 Gy in six fractions over 6 weeks) radiation. The primary endpoint was centered on the OS. While RT improved the failure-free survival (FFS) (HR: 0.76 [95% CI: 0.68–0.84], *p* < 0.0001), it did not yield a significant difference in the OS (HR: 0.92 [95% CI: 0.80–1.06], *p* = 0.266). Notably, among patients with low-volume metastases, the SOC + RT group exhibited a significantly enhanced OS (HR = 0.68 [95% CI: 0.52–0.90], *p* = 0.007) and FFS (HR = 0.59 [95% CI: 0.49–0.72], *p* < 0.0001) [44]. Moreover, the 3-year survival rate was 73% for patients in the SOC group, whereas it escalated to 81% for those in the SOC + RT group [44].

The PEACE-1 trial initially established that combining SOC (ADT with or without docetaxel) with abiraterone acetate plus prednisone (AAP) led to improvements in both the OS and the radiographic PFS (rPFS) among patients with de novo mCSPC [45]. More recently, Bossi et al. reported findings related to prostate irradiation in de novo mCSPC patients featuring low-volume metastases [46]. Surprisingly, RT did not confer an OS advantage in this subset of patients. The median OS stood at 6.9 years (95.1% CI: 5.9–7.5) without RT compared to 7.5 years (6.0-NR) with RT (HR = 0.97 [95% CI: 0.74–1.27], *p* = 0.81). Nevertheless, it is noteworthy that the most favorable outcomes in terms of the rPFS and OS were observed in patients receiving SOC + AAP + RT, although the differences did not reach statistical significance [46].

The STOPCAP study [47], employing a prospective framework for an adaptive meta-analysis that incorporated data from three trials (HORRAD [43], STAMPEDE Arm H [44], and PEACE-1 [45]), examined the utility of primary prostate RT. The analysis revealed that, in unselected patients, primary prostate RT failed to confer significant benefits in terms of the OS (HR = 0.92 [95% CI: 0.81–1.04], *p* = 0.195) or the PFS (HR = 0.94 [95% CI: 0.84–1.05], *p* = 0.238). Nevertheless, an intriguing observation emerged within the subgroup characterized by fewer than five bone metastases, where a notable 7% enhancement in the 3-year survival rates surfaced. This finding lends credence to the proposition of a pertinent role for localized RT in the realm of oligometastatic CSPC.

### 4.4. Metastasis-Directed Treatment

MDT, exemplified by salvage lymph node dissection or SBRT, when employed in the context of oligometastatic PCa, holds the promise of conferring advantages such as the retardation of further metastatic spread and the postponement of ADT initiation. Noteworthy prospective randomized controlled trials that have examined the realm of MDT include the STOMP trial [48] and the ORIORE trial [49].

The STOMP trial, a multicenter phase II study, randomized patients afflicted with oligorecurrent mCSPC. This study focused on asymptomatic PCa patients who had previously undergone definitive therapies, including surgery and/or radiation, and exhibited three or fewer metastases (any N1 or M1), as confirmed by choline PET-CT screening [48]. Among the sixty-two patients, random assignment into either active surveillance or MDT (surgery or SBRT) transpired, with the primary endpoint centered on the ADT-free survival. After a median follow-up of 36 months, those subjected to MDT demonstrated a more favorable ADT-free survival (21 months versus 13 months, HR = 0.60 [95% CI: 0.40–0.90], *p* = 0.11). At the 5-year juncture, the ADT-free survival stood at 34% and 8% for the MDT and surveillance cohorts, respectively [50].

The ORIOLE trial, a prospective phase II randomized controlled trial, encompassed 54 patients grappling with oligorecurrent mCSPC, a diagnosis corroborated through conventional imaging means [49]. Patients underwent random allocation into the SBRT arm or the observation group. The primary outcome measure revolved around the progression at 6 months post-randomization, underpinned by the hypothesis that SBRT targeting all metastases could forestall progression by interrupting the metastatic cascade. After 6 months of follow-up, progression manifested in only 19% of the patients in the SBRT arm, compared to 61% in the control group (*p* = 0.005). SBRT improved the median PFS (not reached versus 5.8 months, HR = 0.30 [95% CI: 0.30–0.81], *p* = 0.002). Notably, SBRT was well tolerated and emerged as a viable option, with minimal acute toxicity observed in this patient cohort [49].

The initial findings from the STOMP and ORIOLE trials indicated that MDT in oligometastatic CSPC patients led to an enhanced treatment efficacy. Additionally, Deek et al. evaluated the potential of a high-risk mutational signature in stratifying the risk associated with the outcomes post-MDT [50]. High-risk mutations were defined as pathogenic somatic mutations occurring within the ATM, BRCA1/2, Rb1, or TP53 genes. Their results revealed that MDT significantly prolongs the median PFS compared to mere observation (pooled HR = 0.44 [95% CI: 0.29–0.66], *p* < 0.001), with the most substantial benefit conferred by MDT observed in patients harboring high-risk mutations (HR, high-risk: 0.05; HR, not high-risk: 0.42; *p*-value for interaction: 0.12). Within the MDT group, individuals lacking high-risk mutations experienced a median PFS of 13.4 months, whereas those with high-risk mutations had a median PFS of 7.5 months (HR = 0.53 [95% CI: 0.25–1.11], *p* = 0.09). The enduring results from the two exclusive randomized trials in oligometastatic CSPC affirmed the sustained clinical advantage of MDT over passive observation. The presence of a high-risk mutational signature may serve as a valuable tool for risk stratification in assessing the treatment outcomes following MDT [50].

Moreover, the PROPE study (NCT03304418) delved into bone oligometastases, wherein SBRT was coupled with a radium-223 treatment for patients grappling with oligorecurrent PCa harboring fewer than six bone metastases (M1b). This investigation delved into the ADT-free survival, with the expectation that favorable findings could obviate the decline in quality of life associated with ADT by favoring SBRT and radium-223 over conventional ADT approaches [15,16].

Beyond the realm of bone metastasis treatment, ongoing clinical trials have shifted their focus toward lymph node oligometastasis. The OLIGOPELVIS trial (NCT02274779), an open-label phase II trial [51], targeted 67 patients contending with oligorecurrent PCa featuring fewer than six pelvic lymph node metastases. These patients underwent 6 months of ADT in conjunction with salvage pelvic high-dose intensity-modulated RT. The primary endpoint was a 2-year PFS defined by two consecutive PSA levels surpassing those at the study’s commencement and/or clinical evidence of progression. Encouragingly, the 2- and 3-year PFS rates reached 81% and 58%, respectively. At the 2- and 3-year junctures, the biochemical relapse-free survival rates reached 58% and 46%, respectively. This study underscored the potential for combined high-dose salvage pelvic RT and ADT to extend tumor control in cases of oligorecurrent pelvic node relapses within PCa, all while minimizing undue toxicity [51].

The STORM study (NCT03569241) homed in on patients grappling with oligorecurrent PCa characterized by fewer than six pelvic lymph node metastases, as diagnosed through PET-CT imaging employing choline, FACBC, or PSMA tracers [52]. All the enrolled patients were subjected to 6 months of ADT, followed by MDT involving SBRT or a salvage lymphadenectomy in Arm A and MDT featuring whole-pelvis RT in Arm B. The primary endpoint was centered on the metastasis-free survival. Should the results from Arm B of the STORM trial markedly outshine those of Arm A and demonstrate a reduced incidence of adverse events, the prospect of MDT with whole-pelvic RT emerging as the standard treatment for lymph node oligorecurrence may gain substantial traction in the future.

With regard to oligoprogressive CRPC, a clinical trial exists to substantiate the efficacy of monotherapy by utilizing MDT through SBRT. Additionally, three randomized clinical trials endeavor to ascertain the effectiveness of a confluence of MDT and ARATs. Notably, NCT02816983 is meticulously scrutinizing the effectiveness of SBRT in addressing metastatic lesions among patients grappling with mCRPC featuring fewer than four oligometastases (N1, M1a/b/c). The paramount endpoints encompass the PSA-PFS at the one-year juncture and the OS at the two-year mark. Should the outcomes of NCT02816983 be found favorable, the utilization of SBRT for managing metastatic lesions may be contemplated, even in the face of disease progression to mCRPC.

The ARTO trial (NCT03449719), a multicenter phase II randomized clinical trial, has been meticulously designed to elucidate the advantages accrued from incorporating SBRT into the treatment regimen involving AAP. This investigation focuses its purview upon oligoprogressive mCRPC patients harboring fewer than four metastases (N1, M1a/b) [53]. Within this trial, Arm A stands to benefit from the amalgamation of AAP and SBRT, while Arm B will receive AAP exclusively. The primary endpoint hinges on the rate of biochemical response (BR), a metric defined by a PSA reduction of at least 50% from the baseline, as measured at the six-month milestone following treatment commencement. Significantly, the BR was discerned in 79.6% of patients, with Arm A surpassing Arm B in this regard (92% versus 68.3%, odds ratio = 5.34 [95% CI: 2.05–13.88], *p* = 0.001). Moreover, the addition of SBRT yielded a noteworthy enhancement in the PFS (HR = 0.35 [95% CI: 0.21–0.57, *p* < 0.001]). The ARTO trial underscores the clinical merit of augmenting first-line AAP treatment with SBRT in the context of mCRPC patients.

Furthermore, the PCS IX study (NCT02685397), which marries SBRT with enzalutamide, is being conducted within a cohort of oligoprogressive mCRPC patients harboring fewer than six lesions (N1, M1a/b/c, with brain and liver metastases excluded). Arm A is being administered enzalutamide, while Arm B is receiving enzalutamide concomitant with SBRT. Post-treatment imaging diagnoses will form the basis for comparing and corroborating the PFS. Additionally, the PILLAR trial (NCT03503344), an undertaking that interlaces SBRT with apalutamide, is directing its attention toward oligoprogressive mCRPC patients who exhibit fewer than six oligometastases (N1, M1a/b/c). In this study, Arm A is being subjected to apalutamide in conjunction with SBRT, whereas Arm B is being treated solely with apalutamide. The principal parameter of interest revolves around the rate of post-treatment PSA levels falling below 0.2 ng/mL, a criterion poised for comparison and validation. In the event that the results arising from these two randomized clinical trials are crystalline and compelling, the prospect of attaining PSA responsiveness and deriving survival advantages may indeed emerge, thereby advocating for the integration of SBRT and novel hormonal agents in addition to AAP, even in the milieu of oligoprogressive mCRPC.

### 4.5. Immunotherapy with Focal Therapy

Presently, clinical trials investigating the synergy between immunotherapy and ADT for de novo oligometastatic CSPC are in progress. In contemplation of the abscopal effect, wherein distant lesions beyond the purview of radiation fields regress or vanish, several trials are scrutinizing the potential of local cryoablation or a combination regimen involving SBRT and cancer immunotherapy in invoking immune responses.

NCT02489357 is dedicated to patients afflicted by de novo oligometastatic CSPC, characterized by fewer than five tumors (M1a/b/c). Here, pembrolizumab is being administered in conjunction with ADT and local cryoablation, with an evaluation focusing on achieving a PSA-nadir of less than 0.6 ng/mL and an assessment of treatment safety. Notably, NCT02489357 posits that cryoablation-induced tumor antigens may galvanize immune responses, enhancing the therapeutic prowess of pembrolizumab. Additionally, in NCT03007732, pembrolizumab is being administered alongside ADT and SBRT to patients confronting de novo oligometastatic CSPC, boasting fewer than four metastases (M1a/b/c). Arm A involves the intratumoral administration of SD-101, a Toll-like receptor 9 (TLR9) agonist, while Arm B serves as a control to evaluate the PSA-nadir + 2.0 ng/mL rate. In NCT03007732, intratumoral SD-101 administration is postulated to incite immune responses, thereby augmenting the therapeutic efficacy of pembrolizumab when coupled with ADT and SBRT. While the efficacy of cancer immunotherapy in the context of prostate cancer has remained relatively constrained, a favorable outcome in these two clinical trials could pave the way for the adoption of pembrolizumab in conjunction with local therapy for de novo oligometastatic CSPC.

## 5. Conclusions

Owing to advancements in diagnostic imaging technology, particularly PSMA-PET, the pathogenesis of oligometastatic PCa is swiftly being elucidated. In the foreseeable future, a consensus on the definition of oligometastasis, the standardization of diagnostic imaging modalities, and the delineation of the number and sites of metastases is anticipated. Based on the burgeoning body of evidence, presently, as is reflected in clinical guidelines, the most auspicious therapeutic modalities for oligometastatic PCa encompass RT targeting the primary tumor for de novo oligometastatic CSPC and RT aimed at metastatic lesions, constituting MDT, for oligorecurrent PCa. An RP for patients grappling with oligometastatic PCa holds the promise of ameliorating local symptoms and complications, such as bleeding, bladder outlet obstruction, and ureteral obstruction. It is hoped that a sequential, multimodal, systemic, local, and MDT approach will significantly enhance the prognosis and outcomes of future patients grappling with oligometastatic PCa.

## Figures and Tables

**Figure 1 cancers-16-00507-f001:**
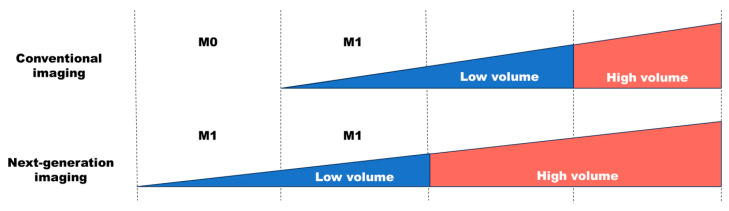
Stage migration owing to next-generation imaging.

**Table 1 cancers-16-00507-t001:** Categorization of oligometastatic PCa.

Category	De Novo Oligometastasis	Oligorecurrence	Oligoprogression
Primary tumor status	Not controlled	Controlled	Controlled/uncontrolled
Systemic treatment	Naive	Naive	Resistant
Location of metastases	N1 or M1	N1 or M1	N1 or M1

PCa: prostate cancer.

**Table 2 cancers-16-00507-t002:** Clinical trials that have included RT for primary or metastatic lesions in oligometastatic PCa.

Study Name	Oligometastasis	Design	Primary Endpoint	Summary of Results
Prostate RT				
HORRAD	De novo mCSPC	ADT vs. ADT+EBRT	OS	Better OS in ADT+EBRT group (not significantly)
STAMPEDE Arm H	De novo mCSPC	SOC (ADT±DOC) vs. SOC+EBRT	OS	Significantly improved OS and FFS in SOC+EBRT group
PEACE-1	De novo mCSPC	SOC (ADT±DOC) ±AAP vs. SOC±AAP +EBRT	OS rPFS	Better OS and rPFS in SOC+AAP+EBRT group (not significantly)
MDT				
STOMP	Oligorecurrent mCSPC	Active surveillance vs. SBRT or surgery for metastases	ADT-free survival	Better ADT-free survival in MDT group (not significantly)
ORIOLE	Oligorecurrent mCSPC	Observation vs. SBRT	rPFS at 6 months	Significantly improved rPFS at 6 months in SBRT group
PROPE	Oligorecurrent mCSPC	SBRT+radium-223	ADT-free survival	-
OLIGOPELVIS	Oligorecurrent mCSPC	ADT+pelvic IMRT	2-year PFS	-
STORM	Oligorecurrent mCSPC	ADT+SBRT or salvage lymphadenectomy vs. ADT+WPRT	MFS	-
ARTO	Oligoprogressive mCRPC	ADT+AAP vs. ADT+AAP+SBRT	BR	Significantly improved BR in ADT+AAP+SBRT group
PCS IX	Oligoprogressive mCRPC	ADT+ENZ vs. ADT+ENZ+SBRT	rPFS	-
PILLAR	Oligoprogressive mCRPC	ADT+APA vs. ADT+APA+SBRT	Post-treatment PSA <0.2 ng/mL	-

AAP: abiraterone acetate plus prednisone, ADT: androgen deprivation therapy, APA: apalutamide, BR: biochemical response, CSPC: metastatic castration-sensitive prostate cancer, mCRPC: metastatic castration-resistant prostate cancer, DOC: docetaxel, EBRT: external beam radiotherapy, ENZ: enzalutamide, FFS: failure-free survival, IMRT: intensity-modulated radiotherapy, MDT: metastasis-directed therapy, MFS: metastasis-free survival, OS: overall survival, PCa: prostate cancer, PSA: prostate-specific antigen, PFS: progression-free survival, rPFS: radiographic progression-free survival, RT: radiotherapy, SBRT: stereotactic body radiation therapy, vs.: versus, WPRT: whole-pelvis radiotherapy.

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
