# Peer review of "Progress in Oligometastatic Prostate Cancer: Emerging Imaging Innovations and Therapeutic Approaches"

_cancers, 2024, doi:10.3390/cancers16030507_

Round 1

Reviewer 1 Report

Comments and Suggestions for Authors

The manuscript is interesting and focuses on diagnostic and therapeutic aspects related to oligometastatic prostate carcinoma. I suggest the authors expand on aspects related to the use of radiopharmaceuticals alternative to PSMA. In particular, 18F-FACBC PET is only briefly mentioned in the text, but no references are provided in this regard. I recommend citing and briefly discussing the following references: DOI: 10.1080/17434440.2022.2117612 and DOI: 10.1007/s00259-019-04506-1. Furthermore, I would emphasize the concept that, at present, a correlation between the identification of oligometastatic disease through PET and improvements in terms of survival has not been clearly demonstrated.

Author Response

Thank you much for you precious comments.

I added the brief topics about 18F-FACBC PET in page 3. Also,  I added your suggested concept that a correlation between the identification of oligometastatic disease through PET and improvements in terms of survival has not been clearly demonstrated in page 4. 

Reviewer 2 Report

Comments and Suggestions for Authors

In this work, the authors have systematically summarized the advancement on oligometastatic prostate cancer study. Like many other review articles published in recent years in this same subject area, this work should be interested to those working in the relevant field and could be served as another information source. The manuscript is well written and the content has included most aspects of oligometastatic prostate cancer, particularly the treatment and imaging analysis were adequately described. Perhaps the pathological mechanisms of the disease, the AR status, and its relationship to current classification (i.e. the concept overlap with castration-resistant prostate cancer) could be described in more details.  

Comments on the Quality of English Language

The phrase in title sentence Advancements in oligometastatic prostate cancer” may need a modification. What the authors described is the advancement on oligometastatic prostate cancer research work (or diagnosis, treatment), not the advancement or progression of the cancer cells.    

Author Response

Thank you so much for your precious comments.

In accordance with your comments, I added the molecular mechanisms of the oligometastatic prostate cancer in page 4, page7-8 and cited the proper references.

Also, I modifed the original title. I changed it to "Progress in oligometastatic prostate cancer: Emerging imaging innovations and therapeutic approaches."

Reviewer 3 Report

Comments and Suggestions for Authors

The manuscript concerns oligometastatic prostate cancer. The topic is clinically relevant, but the paper is poorly prepared.

1. the title is misleading because the paper does not concern what the authors write,

2. the literature review is very poor and does not provide enough new information for the reader,

3. the data presented by the authors are available in many articles and even academic textbooks,

4. tables and figures are of poor quality,

5. the number of references is terrifyingly small.

Author Response

Thank you for your comments.

1. the title is misleading because the paper does not concern what the authors write.

>I changed the original title to "Progress in oligometastatic prostate cancer: Emerging imaging innovations and therapeutic approaches."

2. the literature review is very poor and does not provide enough new information for the reader,

>As noted by other reviewers, I added the topics about imaging (in Page 3 and 4) and molecular mechanisms (in Page 4 and 7-8).

3. the data presented by the authors are available in many articles and even academic textbooks,

>I disagree with that point. This review article contains the most recent data available in the literature.

4. tables and figures are of poor quality,

>I disagree with that point. I dare to use simple table and figure to make it easier for readers to understand.

5. the number of references is terrifyingly small.

>As noted by other reviewers, I cited the reference more.

Round 2

Reviewer 3 Report

Comments and Suggestions for Authors

The research topic was neglected by the authors. The manuscript adds absolutely nothing new to the understanding of the topic. The answers provided by the authors are trivial.

The figure in the manuscript is of poor quality.

There are no tables in the manuscript.

The literature review is very sparse. The number of references is embarrassingly low.

The manuscript contains many generalizations and scientific ambiguities.

Comments on the Quality of English Language

Minor editing of English language required.

Author Response

The research topic was neglected by the authors. The manuscript adds absolutely nothing new to the understanding of the topic. The answers provided by the authors are trivial.

> Our review includes evidence from the most recent papers and conference presentations. Did this reviewer read our paper carefully? Also, this reviewer should peer review our manuscript in detail and present exactly what needs to be modfied.

The figure in the manuscript is of poor quality.

>Our figure is important to help the reader understand.

There are no tables in the manuscript.

>We added new table in accordance with editor's suggestion.

The literature review is very sparse. The number of references is embarrassingly low.

>We cite all important and evident papers. It is better not to cite unnecessarily many papers.

The manuscript contains many generalizations and scientific ambiguities.

>As you pointed out last time, our review article contains solid evidence that is partly included in the guidelines. Are you suggesting that those evidences are ambiguous?

Round 3

Reviewer 3 Report

Comments and Suggestions for Authors

I stand by my previous opinions about the manuscript. The paper is prepared carelessly, with minimal literature. I don't foresee any reader interest in this manuscript or its citation. The work is just academically poor.

The authors did not want to correct the manuscript following my comments, they did not even remove a dot from the title. The replies to my comments are not only trivial but also simply inconsistent with the principles of academic cooperation, they are in a sense rude and inappropriate to the reviewer's comments.

Comments on the Quality of English Language

Moderate editing of the English language is required.